# Self-Reported Pain and Pain Observations in People with Korsakoff’s Syndrome: A Pilot Study

**DOI:** 10.3390/jcm12144681

**Published:** 2023-07-14

**Authors:** Erik Oudman, Thom van der Stadt, Janice R. Bidesie, Jan W. Wijnia, Albert Postma

**Affiliations:** 1Experimental Psychology, Helmholtz Institute, Utrecht University, 3584 CS Utrecht, The Netherlands; 2Slingedael Center of Expertise for Korsakoff Syndrome, Slinge 901, 3086 EZ Rotterdam, The Netherlands

**Keywords:** Korsakoff syndrome, pain, pain manifestations, neuropsychiatric symptoms

## Abstract

Korsakoff’s syndrome (KS) is a chronic neuropsychiatric disorder. The large majority of people with KS experience multiple comorbid health problems, including cardiovascular disease, malignancy, and diabetes mellitus. To our knowledge pain has not been investigated in this population. The aim of this study was to investigate self-reported pain as well as pain behavior observations reported by nursing staff. In total, 38 people diagnosed with KS residing in a long-term care facility for KS participated in this research. The Visual Analogue Scale (VAS), Pain Assessment in Impaired Cognition (PAIC-15), Rotterdam Elderly Pain Observation Scale (REPOS), and the McGill Pain Questionnaire–Dutch Language Version (MPQ-DLV) were used to index self-rated and observational pain in KS. People with KS reported significantly lower pain levels than their healthcare professionals reported for them. The highest pain scores were found on the PAIC-15, specifically on the emotional expression scale. Of importance, the patient pain reports did not correlate with the healthcare pain reports. Moreover, there was a high correlation between neuropsychiatric symptoms and observational pain reports. Specifically, agitation and observational pain reports strongly correlated. In conclusion, people with KS report less pain than their healthcare professionals indicate for them. Moreover, there is a close relationship between neuropsychiatric symptoms and observation-reported pain in people with KS. Our results suggest that pain is possibly underreported by people with KS and should be taken into consideration in treating neuropsychiatric symptoms of KS as a possible underlying cause.

## 1. Introduction

Korsakoff’s syndrome (KS) is a neuropsychiatric disorder caused by a thiamine deficiency [1]. KS is often associated with chronic abuse of alcohol [2]. The prevalence rates of KS are between 1 and 2% in the general population and 12 and 14% in the alcohol misusing population [1]. Due to a degradation of the frontocerebellar circuit (consisting of the executive circuit and motor loops) and the Papez circuit (thalamus, mammillary bodies, and cingulate gyrus), individuals with KS typically exhibit cognitive and motor deficits [3,4]. The main cognitive deficits of KS include anterograde and retrograde amnesia, executive and working memory dysfunction, confabulation, apathy, and affective and social-cognitive impairments [5,6]. Recently, Van Dam et al. [7] reported various behavioral symptoms of emotional or psychological distress such as restlessness, disinhibition, and aggression. These neuropsychiatric symptoms are associated with a higher caregiver burden [8].

Not only do people with KS suffer from cognitive, behavioral, and neuropsychiatric problems, but health problems are also very common in KS [1,7]. Gerridzen and Goossensen [8] showed that over 50% of people with KS in need of residential care had at least one comorbid somatic and/or comorbid psychiatric condition, such as diabetes mellitus, chronic obstructive pulmonary disease (COPD), or depression. It has also been established that people with KS on average only live 5 years after their diagnosis, because of the severity of the comorbid somatic conditions [9,10]. Because people with KS often suffer many comorbid somatic diagnoses, and these somatic comorbidities have a negative effect on health, it is reasonable to expect that people with KS could experience pain [10].

Although the severity of somatic conditions in KS is often complex, clinical experience suggests that people with KS seldomly report pain themselves. Recently, typical pain pathways have been investigated in relation to critical damage in KS, showing that almost all aforementioned pain pathways are likely to be disturbed in KS. More specifically, characteristic lesions to the thalamus, periaqueductal gray, and projections to the cerebellum are all candidate regions for a diminished nociceptive input in other populations, possibly also leading to a diminished pain response in KS [11]. It is therefore relevant to study pain perception in KS.

The perception of pain is regarded as a multidimensional complex experience comprising sensory, affective, and cognitive aspects that can lead to physiological, emotional, and behavioral responses [12]. The pain pathway, the spinothalamic tract, consists of the medial and the lateral pain system [13]. The medial pain system plays a crucial part in the motivational–affective features of pain, involving memory, expectation and emotion, and cognitive–evaluative features, the autonomic–neuroendocrine responses evoked by pain [13,14]. However, the lateral pain system is particularly involved in the sensory–discriminative features of pain such as the recognition of location, intensity, and nature of nociceptive stimuli (e.g., sharp or dull) [14]. It is still unknown which parts of these pain systems are involved in KS, but it has been confirmed in rodents that the periaqueductal gray, cerebellar, and cerebral cortices are involved in increased pain thresholds [11,15,16]. Since these brain areas are also impaired in people with KS, it could be that there might also be higher pain thresholds in KS [11]. In contrast, people with KS could also experience more pain because of axonal polyneuropathy [1].

In other populations diagnosed with neurological or neuropsychiatric disorders, altered pain perception is common [17]. Some researchers found that people with Alzheimer’s dementia may have an increase in the affective component of pain because of a higher pain-related activity in sensory, affective, and cognitive processing regions [18]. In contrast, both increased and decreased pain perception was found in frontotemporal degeneration [19]. In a study of Fletcher et al. [20], there was a decreased pain perception in the behavioral variant of frontotemporal dementia. In these studies, both self-reported pain and proxy-based observational instruments were applied to index pain [21].

Currently it is unknown whether people diagnosed with KS over- or underreport pain. People with KS have serious cognitive- and health-related issues as well as brain damage in critical regions involved in pain perception, suggesting an increased risk for altered pain perception in comparison to a healthy population. Neurological damage in people with KS could results in a diminished ability to feel pain, while polyneuropathy could make them more at risk to feel pain. The aim of the current study was therefore to investigate the perception of pain in people with KS, applying both self-report and observational instruments. Furthermore, we wanted to investigate the relationship between neuropsychiatric symptoms and pain in KS.

## 2. Materials and Methods

### 2.1. Participants

In this study 38 people diagnosed with KS were enrolled. They were all inpatients of Slingedael Korsakoff Expertise Center, Rotterdam, The Netherlands. In total, 31 males and 7 females with a mean age = 65.5 ± 8.07 participated. All people with KS were in the chronic, amnesic phase of the syndrome and were residing in a long-term care facility for KS. Prior to their diagnosis, all patients underwent extensive neuropsychological assessment showing memory problems. All participants had a history of Wernicke’s encephalopathy, as indicated by medical charts. Moreover, all had an extensive history of health issues (bone fractures, heart problems, malignancy, diabetes mellitus, etc.), but the selection for this study was not based on the presence of comorbid diagnoses. All participants were able to read and speak Dutch. The primary responsible nurse was asked to score the observational instruments for the patients after approval of the patient and informant of the patient. They were asked to do so because they knew the patient best and had the most frequent contact with them within the care staff. Participants did not receive financial compensation for their participation. Informed consent and assent was obtained via the patient and legal representative for all patients. Ethical approval was obtained by the faculty ethics committee of the social sciences department of the University of Utrecht, Utrecht, The Netherlands (registration code: 20-072).

### 2.2. Materials

*Subjective pain:* A subjective question about their current pain was asked of both participants and healthcare professionals. People with KS were asked if they could answer yes/no to the question “Do you feel pain at the moment?”; the healthcare professional was asked if he/she could answer yes/no to the question “Do you think the person with KS is in pain at the moment?”. The Visual Analogue Scale (VAS) was used to measure pain in KS patients [22]. VAS is a nonspecific measurement scale, consisting of a horizontal line. The length of this line is 10 cm. On the left side is the minimum score (no pain), on the right side is the maximum score (worst possible pain). The patient were asked to tick perpendicular to the line to what extent they experienced physical pain. The questions that were asked on this scale were: “How severe was your pain on average over the past week (7 days)?” and “How severe was your pain at the worst moments in the past week (7 days)?” The number of millimeters between the line indicated by the patient was used as an outcome. The reliability of the VAS for acute pain measurement appears to be high in other populations [23]. Because a lack of awareness is a central characteristic of KS [24] and to increase the reliability of this study, the VAS-scale was measured twice within approximately two weeks of time. An earlier study suggested a relatively preserved ability to reliably score loneliness in KS patients, suggesting the possibility that also VAS scores could be assessed reliably [25].

*Localization of pain:* To measure the localization of the pain in KS patients, the McGill Pain Questionnaire–Dutch Language Version (MPQ-DLV) was used. The MPQ-DLV is used to measure complaints about the pain [22]. Since several pain questionnaires had already been included in this research, only the section about pain localization was used. This section of the MPQ-DLV consists of a picture of a human body from the front and the back. The patient had to indicate where they felt/experienced the worst pain over the past 7 days by drawing a cross. The test–retest correlations of the nine indices and the visual analog pain intensity scales ranged from 0.62 to 0.93 (median: 0.84) [26]. Cronbach’s alpha coefficients for the indices varied between 0.61 and 0.85 (median: 0.72) [24].

*Pain observational instruments:* The Pain Assessment in Impaired Cognition (PAIC-15) is an observational scale to assess pain in persons with impaired cognition [27]. This observation instrument was filled in by the primary responsible nurse of the patient. The PAIC-15 consists of 15 items divided into three domains each with five items: body movements, vocalizations, and facial expressions. Each item has a title and explanation to avoid ambiguity. Each item is scored on a 0-to-3 scale: 0 = not at all, 1 = slight degree, 2 = moderate degree, and 3 = great degree. Moreover, there is an option “not scorable” for each item. The total PAIC-15 score is calculated by summing all the item scores. The higher the sum, the higher the probability the person is in pain. The 15 items are: frowning, narrowing eyes, raising upper lip, opening mouth, looking tense, freezing, guarding, resisting care, rubbing, restlessness, pain-related words, shouting, groaning, mumbling, and complaining. The inter-rater reliability of the PAIC-15 is very high for all three domains (facial expression: 0.91, vocalization items: 0.93, body movements: 0.92; aggregated kappa across domains: 0.92) in older patients with cognitive disorders [28].

The Rotterdam Elderly Pain Observation Scale (REPOS) was used by the healthcare professionals to assess behaviors associated with pain in KS patients [29]. The REPOS works with an instruction card that describes 10 behaviors that are seen as typical of pain. The observer scores as absent (0) or present (1) after a 2 min observation period. The total scores range from 0 to 10. The REPOS has been determined as a valid and reliable instrument to assess pain in cognitively impaired individuals by several studies [29,30].

*Neuropsychiatric symptoms:* The prevalence of neuropsychiatric problems was measured with the Neuropsychiatric Inventory Questionnaire (NPI-Q) [31]. The NPI-Q is a brief questionnaire form of the NPI that was originally developed for the assessment of 12 domains on behavioral and psychological symptoms that are common in dementia. The Dutch translation used in this study has been demonstrated to be reliable and valid [32]. The primary nurse completed the NPI-Q for each patient in his/her section. The NPI-Q total severity score is the sum of the symptom scores and ranges from 0 to 36. Caregiver distress associated with neuropsychiatric problems was measured with the NPI-Q distress subscale. This subscale of the NPI-Q provides a reliable and valid measure of subjective caregiver distress in relation to neuropsychiatric problems [33]. The caregiver rated the level of distress experienced in relation to one of the 12 symptoms on a 6-point scale ranging from 0 (no distress) to 5 (severely distressed). The total distress score is calculated by summing the distress scores of the individual symptoms and ranges from 0 to 60.

### 2.3. Procedure

Participants were seen twice within a two-week interval. On both occasions the participant was asked to complete the VAS. On the same day, the primary responsible nurse (a healthcare professional) was asked if the patient would feel pain at that moment, with two answer categories: “yes” or “no”. The caregiver was unfamiliar with the answers of the patient. After this question, the primary responsible nurse was asked to fill in the NPI-Q, REPOS, and PAIC-15 for all patients that were enrolled in his/her department of the Korsakoff Center.

### 2.4. Statistical Procedure

For all included variables, means and standard deviations were calculated. We performed Kolmogorov–Smirnov tests for normality. Based on a normality assumption violation, we calculated *U*-tests to evaluate whether there was a significant difference between males and females regarding age, MoCA, REPOS, VAS, PAIC-15, NPI-Severity, and NPI-Distress scores. Pain as indicated by persons with KS and their healthcare professionals was presented in a cross-tabulation of Pearsons’s correlations between pain measurements (VAS week 1, VAS week 2, REPOS, PAIC-15) and cognition (MoCA), and neuropsychiatric symptoms (NPI-Q). For the PAIC-15 and self-reported pain, scores were compared to a reference group of residential patients diagnosed with dementia (n = 197), previously published by Van der Steen et al., 2021 [34]. Statistical differences were calculated with Fisher’s exact test.

## 3. Results

### 3.1. General Overview

In total, 38 patients were included. One male patient was excluded based on missing values in his self-report of pain. The total number of participants of this research was therefore 37 patients, and the general characteristics and pain scores are represented in Table 1. Seven female patients and thirty male patients were included. Male patients had higher levels of self-reported pain than female patients (see Table 1).

### 3.2. Pain Perception

Of the 37 included participants diagnosed with KS, 6 reported pain (16%). According to the healthcare professionals, 11 patients did show signs of pain (29.7%).

### 3.3. Somatic Comorbidities and Pain

In this study, 75.7% of the KS patients had multiple comorbid somatic diagnoses possibly leading to pain. The large majority had a combined respiratory and heart disease, 10.8% of the KS patients only had a respiratory disease. Moreover, 5.4% of the KS patients had a digestive disease. In addition, three patients (8.1%) had diabetes and hypertension. The comorbid conditions possibly leading to pain can be seen in Table 1.

### 3.4. Localization of Pain

To investigate pain localization, participants were asked to indicate whether they felt pain in specific body parts with the MPQ-DLV pain localization scale. Of all participants, 19 did not report any pain in their body (51.4%). In total, eight patients reported pain in their legs (21.6%), three in their stomach (8.1%), three in their head (8.1%), two in their hands (5.4%), one in their back and in their hands (2.7%). None of the patients reported pain in multiple locations. These results indicate that with a body chart, patients report pain, while they do so less frequently without a body chart.

### 3.5. Visual Analogue Scales (Self-Reported Pain)

On both test occasions participants had to indicate on average how much pain they experienced the last seven days. On average, patients reported 2.0 (SD = 2.8) out of 10 on the first occasion, and 2.4 (SD = 3.3) out of 10 on the second occasion. Nineteen participants scored a 0 on the first occasion, and twenty-one participants score a 0 on the second occasion. Seven patients scored higher than a 5 out of 10 on the first occasion (range 5–8.4), and nine on the second occasion (range 5–10). Their worst pain in the last seven days was indicated as a 2.3 (SD = 2.7) on the first occasion and a 2.8 (SD = 2.9) on the second occasion. A strong correlation was found in the first and second week of the first item of the Visual Analogue Scale (r (37) = 0.72, *p* < 0.001). Furthermore, a moderate to high correlation was found for the second item of the Visual Analogue Scale in the first and second week (r (37) = 0.63, *p* < 0.001).

### 3.6. Observation Scales for Pain Signals Observed by Healthcare Professionals

Whether patients reported pain themselves or the primary nurses reported pain for them is represented in Table 2. On the REPOS, 8 out of 37 participants scored a 4 or higher, indicative of possible pain. In total, 21 participants scored a one or higher. The average score was 1.7 (SD: 2.4).

On the PAIC-15, 24 participants scored a 3 or higher, indicative of possible pain. In total, 34 participants score a 1 or higher. The mean score was 8.9 (SD: 9.7). The mean subscale scores were 3.9 (SD: 3.9) for facial expressions of pain (range: 0–15), 2.4 (SD: 3.2) for body movements indicative of pain (range: 0–14), and 2.6 (SD: 3.4) for voice sounds indicative of pain (range: 0–12). In Table 3, the scores on the PAIC-15 with self-reports of pain were compared to a reference group of dementia patients, previously published by Van der Steen et al. (2021). Results suggest that with low PAIC-15 scores, KS patients tend to report more pain themselves than dementia patients. More globally, patient profiles of dementia patients and KS patients are divergent.

### 3.7. Congruency between Self-Observed Pain and Observational Scales

The correlation coefficient between PAIC-15 total scores and VAS total scores (session 1) were not significant (r = −0.2, *p* = 0.17). Moreover, the correlation coefficient between the REPOS total score and VAS total scores (session 1) was not significant (r = 0.02, *p* = 0.28). Both results suggest a limited congruency between self-observed pain and observation-based pain.

### 3.8. Neuropsychiatric Symptoms in Korsakoff Syndrome

For all participants, neuropsychiatric symptoms were evaluated by means of the NPI-Q. The subscale scores are represented in Table 4. The three most severe neuropsychiatric symptoms, irritability, agitation, and apathy, also resulted in the most caregiving distress.

### 3.9. Correlations

Correlations between self-observed pain (VAS-scales), observational scales (PAIC-15 and REPOS), cognition (MoCA), and neuropsychiatric symptoms (NPI-Q) were calculated. Results are displayed in Table 5. A moderate correlation was found between the pain observation scales and neuropsychiatric symptoms.

To scrutinize the relationship between neuropsychiatric symptoms and pain, the top three NPI-Q severity and NPI-Q caregiver distress items were calculated with the PAIC-15 and REPOS. The severity of irritability (r = 0.42), severity of agitation (r = 0.69), and severity of apathy (r = 0.46) all correlated significantly with the PAIC-15 scores. The severity of agitation (r = 0.55) and severity of apathy (r = 0.33) correlated significantly with the REPOS scores.

Caregiver distress resulting from irritability (r = 0.42), agitation (r = 0.70), and apathy (r = 0.36) correlated significantly with the PAIC-15 scores. Caregiver distress resulting from agitation (r = 0.55) correlated significantly with the REPOS scores. Together these results suggest that irritability and agitation are strongly correlated with observational scales for pain.

## 4. Discussion

The aim of this study was to investigate self-reports and observational reports of pain in people with KS. Self-reports of pain by people with KS were low, despite the frequent presence of somatic comorbidities. Self-reports of pain were also lower than reports by healthcare professionals in people with KS. In female patients, self-reports were even lower than in male patients. Although self-reported pain was reliable over time, scores were not congruent with reports from healthcare professionals. Scores on the PAIC-15 were systematically higher than on the REPOS. People with KS displayed facial expressions of pain, and also body movements or voices of pain on the PAIC-15. Both observational instruments for behaviors associated with pain strongly correlated with neuropsychiatric symptoms, most specifically the severity and caregiver distress of agitated behavior possibly suggesting a complex relationship between pain and neuropsychiatric symptoms in KS.

A recent review by our research group suggested largely affected pain pathways in KS, possibly resulting in a compromised ability to perceive pain in KS [11]. These findings are in line with the current study, displaying minimal pain reports of people with KS. The low-pain reports are also in line with earlier studies in patients diagnosed with Alzheimer’s disease and other cognitive disorders [17,18]. Self-reports of pain in KS are likely to underestimate actual pain sensation and behavior in KS, as indicated by the relatively high number of comorbid somatic diagnoses, low number of reports of pain, and lack of congruency with caregiver reports of pain. As a comparison between our data and recently published data [34] on dementia patient suggests, the self-reports of pain are even more divergent from the proxy-based reports of pain in KS compared to dementia patients. More research to elucidate the relationship between self-reported pain and proxy observations of pain is required.

Observational instruments, such as REPOS and PAIC-15, were developed to estimate pain in patients with cognitive difficulties. One important factor in the development of both instruments was the relative underdiagnosis and undertreatment of pain in cognitively compromised adults [17,18]. On both instruments, many people with KS show multiple signals of pain behavior. In comparison to other populations, such as people with Alzheimer’s disease, scores on the PAIC-15 were high. Recently, people with Alzheimer disease scored a 4.6 (SD: 5.2), while people with KS scored an 8.9 (SD: 9.7) in our study [34].On the REPOS, scores were more comparable with earlier results in Alzheimer’s disease [28]. Further research is needed to indicate which observational instrument is most suitable for KS.

One of the most important findings of our study was that neuropsychiatric symptoms strongly correlated with pain observations. Specifically, agitated behavior and the concomitant caregiver distress resulting from this behavior had a strong relationship with observations of pain. Earlier systematic research in people with dementia did not highlight such a strong relationship between pain behaviors and neuropsychiatric symptoms [34]. In people with KS, neuropsychiatric symptoms are a large problem, leading to a high care burden [7,32]. Based on our results it is possible that undiagnosed pain (behavior) is in fact one of the primary factors leading to agitated behaviors in KS. Actively observing and treating pain signals in people with KS is of interest, since the use of psychotropic drugs in KS is relatively high in treating neuropsychiatric problems [8]. Moreover, since the available studies on pain in KS are relatively scarce, it is likely that pain is an overlooked cause of neuropsychiatric symptoms in KS.

Prompting people with KS to actively score pain in specific body parts led to more observed pain than asking whether patients experienced pain without additional prompts. Most often, self-reported leg pain was observed, possibly relating to polyneuropathy that is common in KS [5]. Giving active prompts seems therefore relevant in assessing pain complaints in people with KS, possibly helping compromised memory functioning. Combining self-reported pain complaints and observational scales can give a better indication of pain levels, since the congruency between prior self-reports of pain and observational pain were low.

One of the striking findings in our study was that patients and healthcare reports of pain were not in line, suggesting discrepancies between observed pain behavior by people with KS and precepted pain by the people themselves. Earlier studies on other characteristics in KS found comparable discrepancies, such as reported loneliness [25,35] and quality of life [36]. In the study on quality of life, the extent of the discrepancy between self-report and proxy report was tightly linked to a decreased illness insight of the patients. Since this characteristic is very common in KS patients, a lack of illness insight could also explain the discrepancies between observed behavior associated with pain by people with KS and pain reported by the patients themselves. In future research into pain in KS it would be relevant to include a direct measurement of illness insight.

In our study, we found that males with KS syndrome showed higher self-reports of pain severity than females with KS syndrome. Males were overrepresented in our study, in agreement with three-quarter of the people with KS being male [37]. Because the sample size of the female group was relatively small, the generalization of this finding is possibly limited. Earlier research by El Haj et al. also found evidence of sex differences between males and females diagnosed with KS. In their study, female patients showed a better inhibition behavior than males [38]. It would be of interest to investigate sex differences between people with KS in larger samples.

A strength of the present study is that it is the first study on pain perception in KS, and both self-report and proxy-based reports were included in this study. A limitation of the present study is the lack of multiple assessments of pain and the lack of a control group. Moreover, the sample size was quite small and the population of people with KS is heterogenous. The small sample size could possibly undermine the external and internal validity of this study. Nevertheless, this study points out valuable suggestions for future research on the complex relationship between neuropsychiatric symptoms in KS and pain, and the likeliness of underdiagnosed pain in KS. It is currently unclear whether people with KS have altered pain networks, although a recent study suggests that pain networks are compromised in multiple ways [11]. A suggestion for future studies is to include multiple observations of pain in KS and a reference group. In the current study, we included two assessments of self-reported pain and one assessment of observer-scored pain. Future studies could also incorporate more indices for behaviors associated with pain.

## 5. Recommendations for Clinical Practice

Based on prior neurological evidence suggestive of global lesions in pain pathways in KS [11] and the results of the present study confirming a globally diminished pain response in KS, we recommend clinicians working with KS patients to systematically incorporate pain assessments into the multidisciplinary care for KS patients in (long-term) healthcare facilities. Prompting patients to indicate whether they experience pain in a body part with the help of a body chart (such as the MPQ), as well as the application of a proxy-based pain assessment tool such as the PAIC-15 is recommended to adequately address pain in KS patients.

## 6. Conclusions

To conclude, overall self-reported pain scores by people with KS were relatively low compared to observations of pain by healthcare professionals. Moreover, the convergence between both measurements was low, suggesting that both self-report and proxy-based reports of pain are required to assess pain in people with KS. Importantly, reports of pain were strongly related to neuropsychiatric symptoms in people KS. A possible under-regulation of pain management could therefore be of concern. Additionally, the PAIC-15 can be a useful addition for a better observation of self-reported pain.

## Figures and Tables

**Table 1 jcm-12-04681-t001:** General characteristics and test scores.

Characteristics	Mean (SD) in People with KS (n = 37)	Mean (SD) in Males with KS (n = 30)	Mean (SD) in Females with KS (n = 7)	U-Test Z-Score (*p*-Value)
Age	65.6 (8.1)	66.8 (7.8)	60.6 (7.7)	1.7 (0.08)
MoCA (0–30)	16.9 (4.7)	17.0 (4.4)	16.9 (6.2)	0.3 (0.76)
REPOS (0–10)	1.7 (2.4)	1.4 (1.9)	2.6 (3.8)	0 (1)
VAS (0–10)	2.2 (2.6)	2.7 (2.7)	0.3 (0.5)	2.1 (0.04)
PAIC-15 (0–45)	8.9 (9.7)	8.5 (8.6)	11.4 (13.9)	0.2 (0.86)
NPI-Severity (0–36)	4.9 (5.2)	4.8 (5.2)	4.4 (5.5)	0 (1)
NPI-Distress (0–60)	5.7 (6.8)	5.6 (5.9)	4.9 (6.7)	0.1 (0.92)
Medical comorbidities *Multiple comorbidities* *Respiratory problems* *Digestive system* *Hypertension* *Diabetes* *None*	n= 28 n = 4 n = 2 n = 1 n = 1 n = 1	n = 23 n = 3 n = 1 n = 1 n = 1 n = 1	n = 5 n = 1 n = 1 n = 0 n = 0 n = 0	

SD = standard deviation; MoCA = Montreal Cognitive Assessment; REPOS = Rotterdam Elderly Pain Observation Scale; VAS = Visual Analogue Scale; PAIC-15 = Pain Assessment in Impaired Cognition—15; NPI = Neuropsychiatric Inventory-Q.

**Table 2 jcm-12-04681-t002:** Pain as indicated by persons with KS and their healthcare professionals including heatmap colors (red indicates low scores, green high scores).

Cross-Tabulation for Observed Pain by Caregiver and Patient				
		Pain Reported by Caregiver		Total
		No	Yes	
Pain reported by patient	no	26	5	31
	yes	1	5	6
Total		27	10	37

**Table 3 jcm-12-04681-t003:** Cross-tabulation of PAIC-15 total scores ^a^ against self-report of pain ^b^ in Korsakoff patients and a reference group of patients with dementia in residential care ^c^.

Korsakoff Patients (n = 37 Patients)		
	Reported Pain: Yes or VAS > 0	Reported No Pain and VAS = 0
PAIC-15 ≥ 3 points	11 (30%)	13 (35%)
PAIC-15 < 3 points	8 (22%) *	5 (14%) *
PAIC-15 ≥ 4 points	10 (27%)	12 (32%)
PAIC-15 < 4 points	9 (24%)	6 (16%) *
**Reference group (n = 197 patients)**		
	**Reported pain: yes or VNS > 0**	**Reported no pain and VNS = 0**
PAIC-15 ≥ 3 points	33 (16.8%)	76 (38.6%)
PAIC-15 < 3 points	15 (7.6%)	73 (37.1%)
PAIC-15 ≥ 4 points	25 (12.7%)	57 (28.9%)
PAIC-15 < 4 points	23 (11.7%)	92 (46.7%)

^a^ PAIC-15, Pain Assessment in Impaired Cognition scale. ^b^ Is the patient in pain at this moment (answer yes/no) and how bad is the pain at this moment, according to the VAS, Visual Analogue Scale (0–10) or VNS, Visual Numeric Scale (0–10). ^c^ Patients with Korsakoff syndrome at the Slingedael Korsakoff center. Reference group: patients with dementia in residential care (Van der Steen et al., 2021) [34]. * *p* < 0.05 Fisher’s exact test of number/N and Korsakoff group versus reference group.

**Table 4 jcm-12-04681-t004:** Neuropsychiatric symptoms on the NPI-Q in KS (n = 37).

Scale	Severity	Caregiver Distress
Delusions, mean (SD)	0.3 (0.7)	0.4 (0.9)
Hallucinations, mean (SD)	0.0 (0.0)	0.0 (0.0)
Agitation, mean (SD)	0.7 (0.1)	0.9 (1.5)
Depression, mean (SD)	0.3 (0.7)	0.4 (0.8)
Anxiety, mean (SD)	0.3 (0.6)	0.4 (0.9)
Euphoria, mean (SD)	0.2 (0.4)	0.1 (0.4)
Apathy, mean (SD)	0.7 (1.1)	0.7 (1.2)
Disinhibition, mean (SD)	0.5 (1.1)	0.7 (1.4)
Irritability, mean (SD)	0.9 (1.2)	1.0 (1.4)
Aberrant motor behavior, mean (SD)	0.4 (0.8)	0.4 (1.0)
Nighttime behavior disturbance, mean (SD)	0.2 (0.6)	0.2 (0.9)
Appetite, mean (SD)	0.4 (0.8)	0.4 (1.0)
Total score, mean (SD)	4.9 (5.2)	5.6 (6.8)

**Table 5 jcm-12-04681-t005:** Pearsons’s correlations between pain measurements (VAS, REPOS, PAIC-15), cognition (MoCA), and neuropsychiatric symptoms (NPI-Q).

	VAS	MoCA	REPOS	PAIC-15	NPI- Severity	NPI- Distress
**VAS**	1	0.061	−0.148	−0.125	0.067	0.062
**MoCA**	0.061	1	−0.047	−0.075	−0.215	−0.216
**REPOS**	−0.148	−0.047	1	0.816 **	0.524 **	0.426 **
**PAIC-15**	−0.125	−0.075	0.816 **	1	0.505 **	0.427 **
**NPI-Severity**	0.067	−0.215	0.524 **	0.505 **	1	0.912 **
**NPI-Distress**	0.062	−0.216	0.426 **	0.427	0.912 **	1

VAS = Visual Analogue Scale; REPOS = Rotterdam Elderly Pain Observation Scale; PAIC-15 = Pain Assessment in Impaired Cognition-15; MoCA = Montreal Cognitive Assessment; NPI = Neuropsychiatric Inventory-Q; ** = *p* < 0.01.

## Data Availability

Data is available on inquiry.

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
