# Peer review of "Self-Reported Pain and Pain Observations in People with Korsakoff’s Syndrome: A Pilot Study"

_jcm, 2023, doi:10.3390/jcm12144681_

Round 1

Reviewer 1 Report

Oudman et al., is a nice review on the Korsakoff's syndrome (KS). Since pain has not been investigated before in this population. This study have unraveled the pain perception in Korsakoff's syndrome.

I  have only 1 question

1) Would be possible to show the list of patients with the information of the gender, age and if they live alone. With a sum information of the pain grade for each patient observed by them and observed by the nurse. A simple representation would be a heat map.

Author Response

We now included information on gender distribution in our results section: "Seven female patients and 30 male patients were included".

Information on age can be found in Table 1. 

We included a heat map for Table 2 of the manuscript.

Reviewer 2 Report

Thank you for the opportunity to review this observational study on pain in KS patients. This is an important topic which should definitely be investigated further in the future in order to increase the quality of life for these patients. While the current study definitely has its strengths, I believe there are some issues that could be addressed to further improve the current paper.

Introduction:

-        Please write COPD in full: chronic obstructive pulmonary disease

-        ‘in clinical settings people with KS seldomly report pain themselves’: is this statement based on the literature or own clinical experience of the authors?

Methods:

-        primary responsible nurse did the observational assessment, but was this nurse the nurse that knew the patients best and therefore was the best person to do the assessment?

-        How many of the patients were still capable of giving informed consent themselves?

-        2.2 Materials: “Do you feel pain at that moment?” Shouldn’t this be ‘at the moment’, as ‘that’ refers to a different moment in time?

-        2.2 Materials: The patientS were asked to 115 tick perpendicular to the line to what extent THEY experienceD physical pain

-        2.2 Materials: Is there any information on validity or reliability of using a VAS scale in KS patients, especially patients in the chronic phase of the syndrome? And to what extent were they capable of scoring their pain retrogradely, i.e., pain during the past 7 days? KS patients have trouble with their short term memory.

-        2.2 Materials: localization of pain: “The patient must draw a cross on the body where he/she endures the worst pain.” Do the authors mean that the patients had to indicate where they felt/experienced the worst pain over the past 7 days or at that moment or where they can endure the most pain? This is not exactly clear.

-        Procedure: from the description of the procedure, it is not clear whether the patient and nurse reports (especially the VAS, which is then compared afterwards) were done at the same day or not.

-        Is there any sample size calculation performed before initiating the study?

Results:

-        Did the authors check normality of the data and which tests did you do to assess normality?

-        3.4 Localization of pain: The authors state that “These results indicate that if people with KS are asked to localize pain 214 based on body locations, reports are higher than asked without additional clues.” Can the authors please elaborate? Do you mean that without the body chart they would not report pain anywhere?

-        3.5 Where are the results fort he question ‘Do you feel pain at that moment?’?

Discussion:

-        I miss a discussion on the clinical impact of the findings here, or recommendations towards clinical practice based on the research findings

General: The method section should be revised extensively. Many this are not clearly described and spelling mistakes are part of the explanation for the unclarity.

Some typos cause the method section to be unclear

Author Response

-        Please write COPD in full: chronic obstructive pulmonary disease

Response: We have corrected this error.

-        ‘in clinical settings people with KS seldomly report pain themselves’: is this statement based on the literature or own clinical experience of the authors?

Response: This statement was based on clinical experience of the authors. We therefore changed this sentence to: “Although the severity of somatic conditions in KS is often complex, clinical experience suggests that people with KS seldomly report pain themselves”.

Methods:

-        primary responsible nurse did the observational assessment, but was this nurse the nurse that knew the patients best and therefore was the best person to do the assessment?

Response: yes, the primary responsible nurse was the best person to do the assessment. Unfortunately, family members and (former) friends are often not closely involved in the care for KS patients. We included the following sentence: “They [the responsible nurse] were asked to do so, because they knew the patient best and had the most frequent contact with them within the care staff”.

-        How many of the patients were still capable of giving informed consent themselves?

Response: All patients were still capable of giving informed consent themselves, and for all patients also the legal representative gave informed consent. We have corrected this issue: “Informed consent and assent was obtained via the patient and legal representative for all patients”.

-        2.2 Materials: “Do you feel pain at that moment?” Shouldn’t this be ‘at the moment’, as ‘that’ refers to a different moment in time?

Response: We have corrected this error.

-        2.2 Materials: The patientS were asked to 115 tick perpendicular to the line to what extent THEY experienceD physical pain

Response: We have corrected this error.

-        2.2 Materials: Is there any information on validity or reliability of using a VAS scale in KS patients, especially patients in the chronic phase of the syndrome? And to what extent were they capable of scoring their pain retrogradely, i.e., pain during the past 7 days? KS patients have trouble with their short term memory.

Response: The information on the application of VAS scales in KS is quite limited. We earlier perfomed a study on self-reported loneliness, and currently perform a study on self-reported quality of life. In both projects reliability is high. Regarding validity of self-report information is still limited (and therefore topic of this study). We included an additional text regarding this topic: “Because a lack of awareness is a central characteristic of KS [24] and to increase the reliability of this study, the VAS-scale was measured twice within approximately two weeks of time. An earlier study suggested relatively preserved ability to reliably score loneliness in KS patients, suggesting the possibility that also VAS scores could be assessed reliably [35]”.

-        2.2 Materials: localization of pain: “The patient must draw a cross on the body where he/she endures the worst pain.” Do the authors mean that the patients had to indicate where they felt/experienced the worst pain over the past 7 days or at that moment or where they can endure the most pain? This is not exactly clear.

Response: they had to indicate whether the experienced the worst pain over the past 7 days. We have corrected this typographical error.

-        Procedure: from the description of the procedure, it is not clear whether the patient and nurse reports (especially the VAS, which is then compared afterwards) were done at the same day or not.

Response: we now include the information that they were asked on the same day this in our procedure section.

-        Is there any sample size calculation performed before initiating the study?

Response: The sample size was based on availability of KS patients within the center, not on actual sample size calculations.

Results:

-        Did the authors check normality of the data and which tests did you do to assess normality?

Response: We included a normality analysis (K-S) in the revised manuscript, and changed the t-tests to u-tests based on non-normality for most of the data.

        3.4 Localization of pain: The authors state that “These results indicate that if people with KS are asked to localize pain 214 based on body locations, reports are higher than asked without additional clues.” Can the authors please elaborate? Do you mean that without the body chart they would not report pain anywhere?

Response: We apologize for this inconvenience. Without the body chart they would not report pain. We have corrected this sentence.

-        3.5 Where are the results fort he question ‘Do you feel pain at that moment?’?

Response: We included these findings in Table 2, and also included a reference to Table 1 in section 3.5 of the revised manuscript.

Discussion:

-        I miss a discussion on the clinical impact of the findings here, or recommendations towards clinical practice based on the research findings

Response: We included an additional paragraph on recommendations for clinical practice of the revised manuscript:

  1. Recommendations for clinical practice

Based on prior neurological evidence suggestive for global lesions in pain pathways in KS [11] and the results of the present study confirming a globally diminished pain response in KS, we recommend clinician’s working with KS patients to systematically incorporate pain assessments into multidisciplinary care for KS patients in (long-term) healthcare facilities. Prompting patients to indicate whether they experience pain in a body part with the help of a body chart such as the MPQ, as well as the application of a proxy-based pain assessment tool such as the PAIC-15 is recommended to adequately address pain in KS patients.

General: The method section should be revised extensively. Many this are not clearly described and spelling mistakes are part of the explanation for the unclarity.

Response: We have double checked our method section for unclarities and corrected spelling mistakes in the revised manuscript.

Reviewer 3 Report

The authors discuss pain perception/ detection in KS patients.

The article is well written and describes various aspects of KS that may contribute to pain sensation. The study is of general relevance for patient treatment with KS. I suggest the authors read the manuscript thoroughly to correct minor grammatical mistakes in the manuscript.

Minor Grammatical errors need to be corrected.

Author Response

Thank you. We have corrected typographical mistakes throughout the manuscript.

Round 2

Reviewer 2 Report

I believe the authors addressed my comments well and I think the manuscript has improved sufficiently.